# Avoiding Imposters and Delinquents: Adversarial Crowdsourcing and Peer Prediction

**Jacob Steinhardt**
Stanford University

**Gregory Valiant**
Stanford University

**Moses Charikar**
Stanford University

## Abstract

We consider a crowdsourcing model in which $n$ workers are asked to rate the quality of $n$ items previously generated by other workers. An unknown set of $\alpha n$ workers generate reliable ratings, while the remaining workers may behave arbitrarily and possibly adversarially. The manager of the experiment can also manually evaluate the quality of a small number of items, and wishes to curate together almost all of the high-quality items with at most an $\epsilon$ fraction of low-quality items. Perhaps surprisingly, we show that this is possible with an amount of work required of the manager, and each worker, that does not scale with $n$: the dataset can be curated with $\tilde{\mathcal{O}}\left(\frac{1}{\beta\alpha^3\epsilon^4}\right)$ ratings per worker, and $\tilde{\mathcal{O}}\left(\frac{1}{\beta\epsilon^2}\right)$ ratings by the manager, where $\beta$ is the fraction of high-quality items. Our results extend to the more general setting of peer prediction, including peer grading in online classrooms.

## 1 Introduction

How can we reliably obtain information from humans, given that the humans themselves are unreliable, and might even have incentives to mislead us? Versions of this question arise in crowdsourcing (Vuurens et al., 2011), collaborative knowledge generation (Priedhorsky et al., 2007), peer grading in online classrooms (Piech et al., 2013; Kulkarni et al., 2015), aggregation of customer reviews (Harmon, 2004), and the generation/curation of large datasets (Deng et al., 2009). A key challenge is to ensure high information quality despite the fact that many people interacting with the system may be unreliable or even adversarial. This is particularly relevant when raters have an incentive to collude and cheat as in the setting of peer grading, as well as for reviews on sites like Amazon and Yelp, where artists and firms are incentivized to manufacture positive reviews for their own products and negative reviews for their rivals (Harmon, 2004; Mayzlin et al., 2012).

One approach to ensuring quality is to use *gold sets* — questions where the answer is known, which can be used to assess reliability on unknown questions. However, this is overly constraining — it does not make sense for open-ended tasks such as knowledge generation on wikipedia, nor even for crowdsourcing tasks such as "translate this paragraph" or "draw an interesting picture" where there are different equally good answers. This approach may also fail in settings, such as peer grading in massive online open courses, where students might collude to inflate their grades.

In this work, we consider the challenge of using crowdsourced human ratings to accurately and efficiently evaluate a large dataset of content. In some settings, such as peer grading, the end goal is to obtain the accurate evaluation of each datum; in other settings, such as the curation of a large dataset, accurate evaluations could be leveraged to select a high-quality subset of a larger set of variable-quality (perhaps crowd-generated) data.

There are several confounding difficulties that arise in extracting accurate evaluations. First, many raters may be unreliable and give evaluations that are uncorrelated with the actual item quality; second, some reliable raters might be harsher or more lenient than others; third, some items may be harder to evaluate than others and so error rates could vary from item to item, even among reliable

raters; finally, some raters may even collude or want to hack the system. This raises the question: can we obtain information from the reliable raters, without knowing who they are a priori?

In this work, we answer this question in the affirmative, under surprisingly weak assumptions:

- We do not assume that the majority of workers are reliable.
- We do not assume that the unreliable workers conform to any statistical model; they could behave fully adversarially, in collusion with each other and with full knowledge of how the reliable workers behave.
- We do not assume that the reliable worker ratings match the true ratings, but only that they are "approximately monotonic" in the true ratings, in a sense that will be formalized later.
- We do not assume that there is a "gold set" of items with known ratings presented to each user (as an adversary could identify and exploit this). Instead, we rely on a small number of reliable judgments on randomly selected items, obtained after the workers submit their own ratings; in practice, these could be obtained by rating those items oneself.

For concreteness, we describe a simple formalization of the crowdsourcing setting (our actual results hold in a more general setting). We imagine that we are the dataset curator, so that "us" and "ourselves" refers in general to whoever is curating the data. There are $n$ raters and $m$ items to evaluate, which have an unknown quality level in $[0, 1]$. At least $\alpha n$ workers are "reliable" in that their judgments match our own in expectation, and they make independent errors. We assign each worker to evaluate at most $k$ randomly selected items. In addition, we ourselves judge $k_0$ items. Our goal is to recover the $\beta$-quantile: the set $T^*$ of the $\beta m$ highest-quality items. Our main result implies the following:

**Theorem 1.** *In the setting above, suppose $n = m$. Then there is $k = \mathcal{O}(\frac{1}{\beta\alpha^3\epsilon^4})$, and $k_0 = \tilde{\mathcal{O}}(\frac{1}{\beta\epsilon^2})$ such that, with probability $99\%$, we can identify $\beta m$ items with average quality only $\epsilon$ worse than $T^*$.*

Interestingly, the amount of work that each worker (and we ourselves) has to do does not grow with $n$; it depends only on the fraction $\alpha$ of reliable workers and the desired accuracy $\epsilon$. While the number of evaluations $k$ for each worker is likely not optimal, we note that the amount of work $k_0$ required of us is close to optimal: for $\alpha \leq \beta$, it is information theoretically necessary for us to evaluate $\Omega(1/\beta\epsilon^2)$ items, via a reduction to estimating noisy coin flips.

Why is it necessary to include some of our own ratings? If we did not, and $\alpha < \frac{1}{2}$, then an adversary could create a set of dishonest raters that were identical to the reliable raters except with the item indices permuted by a random permutation of $\{1, \ldots, m\}$. In this case, there is no way to distinguish the honest from the dishonest raters except by breaking the symmetry with our own ratings.

Our main result holds in a considerably more general setting where we require a weaker form of inter-rater agreement — for example, our results hold even if some of the reliable raters are harsher than others, as long as the expected ratings induce approximately the same ranking. The focus on quantiles rather than raw ratings is what enables this. Note that once we estimate the quantiles, we can approximately recover the ratings by evaluating a few items in each quantile.

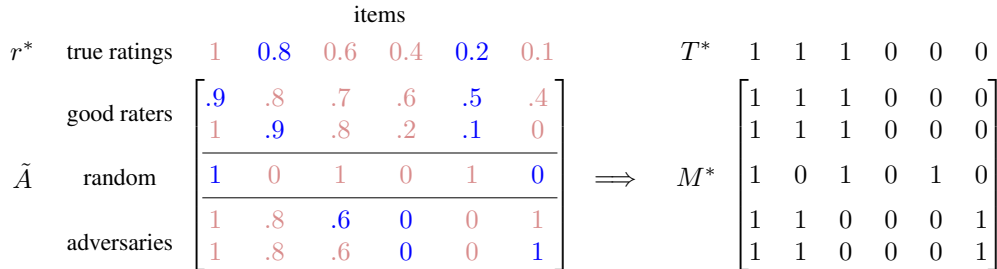

Figure 1: Illustration of our problem setting. We observe a small number of ratings from each rater (indicated in blue), which we represent as entries in a matrix $\tilde{A}$ (unobserved ratings in red, treated as zero by our algorithm). There is also a true rating $r^*$ that we would assign to each item; by rating some items ourself, we observe some entries of $r^*$ (also in blue). Our goal is to recover the set $T^*$ representing the top $\beta$ fraction of items under $r^*$. As an intermediate step, we approximately recover a matrix $M^*$ that indicates the top items for each individual rater.

Our technical tools draw on semidefinite programming methods for matrix completion, which have been used to study graph clustering as well as community detection in the stochastic block model (Holland et al., 1983; Condon and Karp, 2001). Our setting corresponds to the sparse case of graphs with constant degree, which has recently seen great interest (Decelle et al., 2011; Mossel et al., 2012; 2013b;a; Massoulié, 2014; Guédon and Vershynin, 2014; Mossel et al., 2015; Chin et al., 2015; Abbe and Sandon, 2015a;b; Makarychev et al., 2015). Makarychev et al. (2015) in particular provide an algorithm that is robust to adversarial perturbations, but only if the perturbation has size $o(n)$; see also Cai and Li (2015) for robustness results when the degree of the graph is logarithmic.

Several authors have considered semirandom settings for graph clustering, which allow for some types of adversarial behavior (Feige and Krauthgamer, 2000; Feige and Kilian, 2001; Coja-Oghlan, 2004; Krivelevich and Vilenchik, 2006; Coja-Oghlan, 2007; Makarychev et al., 2012; Chen et al., 2014; Guédon and Vershynin, 2014; Moitra et al., 2015; Agarwal et al., 2015). In our setting, these semirandom models are unsuitable because they rule out important types of strategic behavior, such as an adversary rating some items accurately to gain credibility. By allowing arbitrary behavior from the adversary, we face a key technical challenge: while previous analyses consider errors relative to a ground truth clustering, in our setting the ground truth only exists for rows of the matrix corresponding to reliable raters, while the remaining rows could behave arbitrarily even in the limit where all ratings are observed. This necessitates a more careful analysis, which helps to clarify what properties of a clustering are truly necessary for identifying it.

## 2 Algorithm and Intuition

We now describe our recovery algorithm. To fix notation, we assume that there are $n$ raters and $m$ items, and that we observe a matrix $\tilde{A} \in [0,1]^{n \times m}$: $\tilde{A}_{ij} = 0$ if rater $i$ does not rate item $j$, and otherwise $\tilde{A}_{ij}$ is the assigned rating, which takes values in $[0,1]$. In the settings we care about $\tilde{A}$ is very sparse — each rater only rates a few items. Remember that our goal is to recover the $\beta$-quantile $T^*$ of the best items according to our own rating.

Our algorithm is based on the following intuition: the reliable raters must (approximately) agree on the ranking of items, and so if we can cluster the rows of $\tilde{A}$ appropriately, then the reliable raters should form a single very large cluster (of size $\alpha n$). There can be at most $\frac{1}{\alpha}$ disjoint clusters of this size, and so we can manually check the accuracy of each large cluster (by checking agreement with our own rating on a few randomly selected items) and then choose the best one.

One major challenge in using the clustering intuition is the sparsity of $\tilde{A}$: any two rows of $\tilde{A}$ will almost certainly have no ratings in common, so we must exploit the global structure of $\tilde{A}$ to discover clusters, rather than using pairwise comparisons of rows. The key is to view our problem as a form of *noisy matrix completion* — we imagine a matrix $A^*$ in which all the ratings have been filled in and all noise from individual ratings has been removed. We define a matrix $M^*$ that indicates the top $\beta m$ items in each row of $A^*$: $M^*_{ij} = 1$ if item $j$ has one of the top $\beta m$ ratings from rater $i$, and $M^*_{ij} = 0$ otherwise (this differs from the actual definition of $M^*$ given in Section 4, but is the same in spirit). If we could recover $M^*$, we would be close to obtaining the clustering we wanted.

---

**Algorithm 1** Algorithm for recovering $\beta$-quantile matrix $\tilde{M}$ using (unreliable) ratings $\tilde{A}$.

---

1: Parameters: reliable fraction $\alpha$, quantile $\beta$, tolerance $\epsilon$, number of raters $n$, number of items $m$
2: Input: noisy rating matrix $\tilde{A}$
3: Let $\tilde{M}$ be the solution of the optimization problem (1):

$$\text{maximize } \langle \tilde{A}, M \rangle, \tag{1}$$
$$\text{subject to } 0 \leq M_{ij} \leq 1 \ \forall i, j,$$
$$\sum_j M_{ij} \leq \beta m \ \forall j, \qquad \|M\|_* \leq \frac{2}{\alpha \epsilon} \sqrt{\alpha \beta n m},$$

where $\| \cdot \|_*$ denotes nuclear norm.
4: Output $\tilde{M}$.

---

**Algorithm 2** Algorithm for recovering an accurate $\beta$-quantile $T$ from the $\beta$-quantile matrix $\tilde{M}$.

1: Parameters: tolerance $\epsilon$, reliable fraction $\alpha$
2: Input: matrix $\tilde{M}$ of approximate $\beta$-quantiles, noisy ratings $\tilde{r}$
3: Select $2\log(2/\delta)/\alpha$ indices $i \in [n]$ at random.
4: Let $i^*$ be the index among these for which $\langle \tilde{M}_i, \tilde{r} \rangle$ is largest, and let $T_0 \leftarrow \tilde{M}_{i^*}$. $\triangleright T_0 \in [0,1]^m$
5: **do** $T \leftarrow$ RANDOMIZEDROUND$(T_0)$ **while** $\langle T - T_0, \tilde{r} \rangle < -\frac{\epsilon}{4}\beta k$
6: **return** $T$ $\triangleright T \in \{0,1\}^m$

The key observation that allows us to approximate $M^*$ given only the noisy, incomplete $\tilde{A}$ is that $M^*$ *has low-rank structure*: since all of the reliable raters agree with each other, their rows in $M^*$ are all identical, and so there is an $(\alpha n) \times m$ submatrix of $M^*$ with rank 1. This inspires the low-rank matrix completion algorithm for recovering $\tilde{M}$ given in Algorithm 1. Each row of $M$ is constrained to have sum at most $\beta m$, and $M$ as a whole is constrained to have nuclear norm $\|M\|_*$ at most $\frac{2}{\alpha\epsilon}\sqrt{\alpha\beta nm}$. Recall that the *nuclear norm* is the sum of the singular values of $M$; in the same way that the $\ell^1$-norm is a convex surrogate for the $\ell^0$-norm, the nuclear norm acts as a convex surrogate for the rank of $M$ (i.e., number of non-zero singular values). The optimization problem (1) therefore chooses a set of $\beta m$ items in each row to maximize the corresponding values in $\tilde{A}$, while constraining the item sets to have low rank (where low rank is relaxed to low nuclear norm to obtain a convex problem). This low-rank constraint acts as a strong regularizer that quenches the noise in $\tilde{A}$.

Once we have recovered $\tilde{M}$ using Algorithm 1, it remains to recover a specific set $T$ that approximates the $\beta$-quantile according to our ratings. Algorithm 2 provides a recipe for doing so: first, rate $k_0$ items at random, obtaining the vector $\tilde{r}$: $\tilde{r}_j = 0$ if we did not rate item $j$, and otherwise $\tilde{r}_j$ is the (possibly noisy) rating that we assign to item $j$. Next, score each row $\tilde{M}_i$ based on the noisy ratings $\sum_j \tilde{M}_{ij}\tilde{r}_j$, and let $T_0$ be the highest-scoring $\tilde{M}_i$ among $\mathcal{O}(\log(2/\delta)/\alpha)$ randomly selected $i$. Finally, randomly round the vector $T_0 \in [0,1]^m$ to a discrete vector $T \in \{0,1\}^m$, and treat $T$ as the indicator function of a set approximating the $\beta$-quantile (see Section 5 for details of the rounding algorithm).

In summary, given a noisy rating matrix $\tilde{A}$, we will first run Algorithm 1 to recover a $\beta$-quantile matrix $\tilde{M}$ for each rater, and then run Algorithm 2 to recover our personal $\beta$-quantile using $\tilde{M}$.

**Possible attacks by adversaries.** In our algorithm, the adversaries can influence $\tilde{M}_i$ for reliable raters $i$ via the nuclear norm constraint (note that the other constraints are separable across rows). This makes sense because the nuclear norm is what causes us to pool global structure across raters (and thus potentially pool bad information). In order to limit this influence, the constraint on the nuclear norm is weaker than is typical by a factor of $\frac{2}{\epsilon}$; it is not clear to us whether this is actually necessary or due to a loose analysis.

The constraint $\sum_j M_{ij} \leq \beta m$ is not typical in the literature. For instance, (Chen et al., 2014) place no constraint on the sum of each row in $M$ (they instead normalize $\tilde{M}$ to lie in $[-1,1]^{n \times m}$, which recovers the items with positive rating rather than the $\beta$-quantile). Our row normalization constraint prevents an attack in which a spammer rates a random subset of items as high as possible and rates the remaining items as low as possible. If the actual set of high-quality items has density much smaller than 50%, then the spammer gains undue influence relative to honest raters that only rate e.g. 10% of the items highly. Normalizing $M$ to have a fixed row sum prevents this; see Section B for details.

## 3 Assumptions and Approach

We now state our assumptions more formally, state the general form of our results, and outline the key ingredients of the proof. In our setting, we can query a rater $i \in [n]$ and item $j \in [m]$ to obtain a rating $\tilde{A}_{ij} \in [0,1]$. Let $r^* \in [0,1]^m$ denote the vector of true ratings of the items. We can also query an item $j$ (by rating it ourself) to obtain a noisy rating $\tilde{r}_j$ such that $\mathbf{E}[\tilde{r}_j] = r_j^*$.

Let $\mathcal{C} \subseteq [n]$ be the set of reliable raters, where $|\mathcal{C}| \geq \alpha n$. Our main assumption is that the reliable raters make independent errors:

**Assumption 1** (Independence)**.** *When we query a pair $(i,j)$ with $i \in \mathcal{C}$, we obtain an output $\tilde{A}_{ij}$ whose value is independent of all of the other queries so far. Similarly, when we query an item $j$, we obtain an output $\tilde{r}_j$ that is independent of all of the other queries so far.*

---

**Algorithm 3** Algorithm for obtaining (unreliable) ratings matrix $\tilde{A}$ and noisy ratings $\tilde{r}, \tilde{r}'$.

1: Input: number of raters $n$, number of items $m$, and number of ratings $k$ and $k_0$.
2: Initially assign each rater to each item independently with probability $k/m$.
3: For each rater with more than $2k$ items, arbitrarily unassign items until there are $2k$ remaining.
4: For each item with more than $2k$ raters, arbitrarily unassign raters until there are $2k$ remaining.
5: Have the raters submit ratings of their assigned items, and let $\tilde{A}$ denote the resulting matrix of ratings with missing entries fill in with zeros.
6: Generate $\tilde{r}$ by rating items with probability $\frac{k_0}{m}$ (fill in missing entries with zeros)
7: Output $\tilde{A}, \tilde{r}$

---

Note that Assumption 1 allows the unreliable ratings to depend on all previous ratings and also allows arbitrary collusion among the unreliable raters. In our algorithm, we will generate our own ratings after querying everyone else, which ensures that at least $\tilde{r}$ is independent of the adversaries.

We need a way to formalize the idea that the reliable raters agree with us. To this end, for $i \in \mathcal{C}$ let $A_{ij}^* = \mathbf{E}[\tilde{A}_{ij}]$ be the expected rating that rater $i$ assigns to item $j$. We want $A^*$ to be roughly increasing in $r^*$:

**Definition 1** (Monotonic raters). *We say that the reliable raters are $(L, \epsilon)$-monotonic if*

$$r_j^* - r_{j'}^* \le L \cdot (A_{ij}^* - A_{ij'}^*) + \epsilon \tag{2}$$

*whenever $r_j^* \ge r_{j'}^*$, for all $i \in \mathcal{C}$ and all $j, j' \in [m]$.*

The $(L, \epsilon)$-monotonicity property says that if we think that one item is substantially better than another item, the reliable raters should think so as well. As an example, suppose that our own ratings are binary ($r_j^* \in \{0, 1\}$) and that each rating $\tilde{A}_{i,j}$ matches $r_j^*$ with probability $\frac{3}{5}$. Then $A_{i,j}^* = \frac{2}{5} + \frac{1}{5} r_j^*$, and hence the ratings are $(5, 0)$-monotonic. In general, the monotonicity property is fairly mild — if the reliable ratings are not $(L, \epsilon)$-monotonic, it is not clear that they should even be called reliable!

**Algorithm for collecting ratings.** Under the model given in Assumption 1, our algorithm for collecting ratings is given in Algorithm 3. Given integers $k$ and $k_0$, Algorithm 3 assigns each rater at most $2k$ ratings, and assigns us $k_0$ ratings in expectation. The output is a noisy rating matrix $\tilde{A} \in [0, 1]^{n \times m}$ as well as a noisy rating vector $\tilde{r} \in [0, 1]^m$. Our main result states that we can use $\tilde{A}$ and $\tilde{r}$ to estimate the $\beta$-quantile $T^*$; throughout we will assume that $m$ is at least $n$.

**Theorem 2.** *Let $m \ge n$. Suppose that Assumption 1 holds, that the reliable raters are $(L, \epsilon_0)$-monotonic, and that we run Algorithm 3 to obtain noisy ratings. Then there is $k = \mathcal{O}\left(\frac{\log^3(2/\delta)}{\beta \alpha^3 \epsilon^4} \frac{m}{n}\right)$ and $k_0 = \mathcal{O}\left(\frac{\log(2/\alpha\beta\epsilon\delta)}{\beta\epsilon^2}\right)$ such that, with probability $1 - \delta$, Algorithms 1 and 2 output a set $T$ with*

$$\frac{1}{\beta m} \left( \sum_{j \in T^*} r_j^* - \sum_{j \in T} r_j^* \right) \le (2L + 1) \cdot \epsilon + 2\epsilon_0. \tag{3}$$

Note that the amount of work for the raters scales as $\frac{m}{n}$. Some dependence on $\frac{m}{n}$ is necessary, since we need to make sure that every item gets rated at least once.

The proof of Theorem 2 can be split into two parts: analyzing Algorithm 1 (Section 4), and analyzing Algorithm 2 (Section 5). At a high level, analyzing Algorithm 1 involves showing that the nuclear norm constraint in (1) imparts sufficient noise robustness while not allowing the adversary too much influence over the reliable rows of $\tilde{M}$. Analyzing Algorithm 2 is far more straightforward, and requires only standard concentration inequalities and a standard randomized rounding idea (though the latter is perhaps not well-known, so we will explain it briefly in Section 5).

## 4  Recovering $\tilde{M}$ (Algorithm 1)

The goal of this section is to show that solving the optimization problem (1) recovers a matrix $\tilde{M}$ that approximates the $\beta$-quantile of $r^*$ in the following sense:

**Proposition 1.** *Under the conditions of Theorem 2 and the corresponding values of $k$ and $k_0$, Algorithm 1 outputs a matrix $\tilde{M}$ satisfying*

$$\frac{1}{|\mathcal{C}|}\frac{1}{\beta m}\sum_{i\in\mathcal{C}}\sum_{j\in[m]}(T_j^* - \tilde{M}_{ij})A_{ij}^* \leq \epsilon \tag{4}$$

*with probability $1 - \delta$, where $T_j^* = 1$ if $j$ lies in the $\beta$-quantile of $r^*$, and is 0 otherwise.*

Proposition 1 says that the row $\tilde{M}_i$ is good according to rater $i$'s ratings $A_i^*$. Note that $(L, \epsilon_0)$-monotonicity then implies that $\tilde{M}_i$ is also good according to $r^*$. In particular (see A.2 for details)

$$\frac{1}{|\mathcal{C}|}\frac{1}{\beta m}\sum_{i\in\mathcal{C}}\sum_{j\in[m]}(T_j^* - \tilde{M}_{ij})r_j^* \leq L\cdot\frac{1}{|\mathcal{C}|}\frac{1}{\beta m}\sum_{i\in\mathcal{C}}\sum_{j\in[m]}(T_j^* - \tilde{M}_{ij})A_{ij}^* + \epsilon_0 \leq L\cdot\epsilon + \epsilon_0. \tag{5}$$

Proving Proposition 1 involves two major steps: showing (a) that the nuclear norm constraint in (1) imparts noise-robustness, and (b) that the constraint does not allow the adversaries to influence $\tilde{M}_{\mathcal{C}}$ too much. (For a matrix $X$ we let $X_{\mathcal{C}}$ denote the rows indexed by $\mathcal{C}$ and $X_{\overline{\mathcal{C}}}$ the remaining rows.)

In a bit more detail, if we let $M^*$ denote the "ideal" value of $\tilde{M}$, and $B$ denote a denoised version of $\tilde{A}$, we first show (Lemma 1) that $\langle B, \tilde{M} - M^*\rangle \geq -\epsilon'$ for some $\epsilon'$ determined below. This is established via the matrix concentration inequalities in Le et al. (2015). Lemma 1 would already suffice for standard approaches (e.g., Guédon and Vershynin, 2014), but in our case we must grapple with the issue that the rows of $B$ could be arbitrary outside of $\mathcal{C}$, and hence closeness according to $B$ may not imply actual closeness between $\tilde{M}$ and $M^*$. Our main technical contribution, Lemma 2, shows that $\langle B_{\mathcal{C}}, \tilde{M}_{\mathcal{C}} - M_{\mathcal{C}}^*\rangle \geq \langle B, \tilde{M} - M^*\rangle - \epsilon'$; that is, *closeness according to $B$ implies closeness according to $B_{\mathcal{C}}$*. We can then restrict attention to the reliable raters, and obtain Proposition 1.

**Part 1: noise-robustness.** Let $B$ be the matrix satisfying $B_{\mathcal{C}} = \frac{k}{m}A_{\mathcal{C}}^*$, $B_{\overline{\mathcal{C}}} = \tilde{A}_{\overline{\mathcal{C}}}$, which denoises $\tilde{A}$ on $\mathcal{C}$. The scaling $\frac{k}{m}$ is chosen so that $\mathbf{E}[\tilde{A}_{\mathcal{C}}] \approx B_{\mathcal{C}}$. Also define $R \in \mathbf{R}^{n\times m}$ by $R_{ij} = T_j^*$.

Ideally, we would like to have $M_{\mathcal{C}} = R_{\mathcal{C}}$, i.e., $M$ matches $T^*$ on all the rows of $\mathcal{C}$. In light of this, we will let $M^*$ be the solution to the following "corrected" program, which we don't have access to (since it involves knowledge of $A^*$ and $\mathcal{C}$), but which will be useful for analysis purposes:

$$\text{maximize } \langle B, M\rangle, \tag{6}$$
$$\text{subject to } M_{\mathcal{C}} = R_{\mathcal{C}}, \qquad\qquad 0 \leq M_{ij} \leq 1 \ \forall i,j,$$
$$\textstyle\sum_j M_{ij} \leq \beta m \ \forall i, \qquad \|M\|_* \leq \frac{2}{\alpha\epsilon}\sqrt{\alpha\beta nm}$$

Importantly, (6) enforces $M_{ij}^* = T_j^*$ for all $i \in \mathcal{C}$. Lemma 1 shows that $\tilde{M}$ is "close" to $M^*$:

**Lemma 1.** *Let $m \geq n$. Suppose that Assumption 1 holds. Then there is a $k = \mathcal{O}\left(\frac{\log^3(2/\delta)}{\beta\alpha^3\epsilon^4}\frac{m}{n}\right)$ such that the solution $\tilde{M}$ to (1) performs nearly as well as $M^*$ under $B$; specifically, with probability $1 - \delta$,*

$$\langle B, \tilde{M}\rangle \geq \langle B, M^*\rangle - \epsilon\alpha\beta kn. \tag{7}$$

Note that $\tilde{M}$ is not necessarily feasible for (6), because of the constraint $M_{\mathcal{C}} = R_{\mathcal{C}}$; Lemma 1 merely asserts that $\tilde{M}$ approximates $M^*$ in objective value. The proof of Lemma 1, given in Section A.3, primarily involves establishing a *uniform deviation result*; if we let $\mathcal{P}$ denote the feasible set for (1), then we wish to show that $|\langle \tilde{A} - B, M\rangle| \leq \frac{1}{2}\epsilon\alpha\beta kn$ for all $M \in \mathcal{P}$. This would imply that the objectives of (1) and (6) are essentially identical, and so optimizing one also optimizes the other.

Using the inequality $|\langle \tilde{A} - B, M\rangle| \leq \|\tilde{A} - B\|_{\text{op}}\|M\|_*$, where $\|\cdot\|_{\text{op}}$ denotes operator norm, it suffices to establish a matrix concentration inequality bounding $\|\tilde{A} - B\|_{\text{op}}$. This bound follows from the general matrix concentration result of Le et al. (2015), stated in Section A.1.

**Part 2: bounding the influence of adversaries.** We next show that the nuclear norm constraint does not give the adversaries too much influence over the de-noised program (6); this is the most novel aspect of our argument.

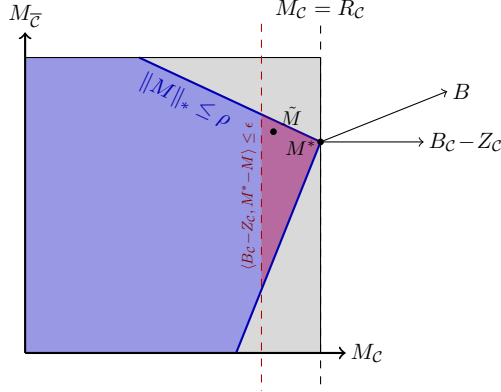

Figure 2: Illustration of our Lagrangian duality argument, and of the role of $Z$. The blue region represents the nuclear norm constraint and the gray region the remaining constraints. Where the blue region slopes downwards, a decrease in $M_{\mathcal{C}}$ can be offset by an increase in $M_{\overline{\mathcal{C}}}$ when measuring $\langle B, M \rangle$. By linearizing the nuclear norm constraint, the vector $B - Z$ accounts for this offset, and the red region represents the constraint $\langle B_{\mathcal{C}} - Z_{\mathcal{C}}, M_{\mathcal{C}}^* - M_{\mathcal{C}} \rangle \leq \epsilon$, which will contain $\tilde{M}$.

Suppose that the constraint on $\|M\|_*$ were not present in (6). Then the adversaries would have no influence on $M_{\mathcal{C}}^*$, because all the remaining constraints in (6) are separable across rows. How can we quantify the effect of this nuclear norm constraint? We exploit Lagrangian duality, which allows us to replace constraints with appropriate modifications to the objective function.

To gain some intuition, consider Figure 2. The key is that the Lagrange multiplier $Z_{\mathcal{C}}$ can bound the amount that $\langle B, M \rangle$ can increase due to changing $M$ outside of $\mathcal{C}$. If we formalize this and analyze $Z$ in detail, we obtain the following result:

**Lemma 2.** *Let $m \geq n$. Then there is a $k = \mathcal{O}\left( \frac{\log^3 (2/\delta)}{\alpha \beta \epsilon^2} \frac{m}{n} \right)$ such that, with probability at least $1 - \delta$, there exists a matrix $Z$ with $\mathrm{rank}(Z) = 1$, $\|Z\|_F \leq \epsilon k \sqrt{\alpha \beta n / m}$, and*

$$\langle B_{\mathcal{C}} - Z_{\mathcal{C}}, M_{\mathcal{C}}^* - M_{\mathcal{C}} \rangle \leq \langle B, M^* - M \rangle \text{ for all } M \in \mathcal{P}. \tag{8}$$

By localizing $\langle B, M^* - M \rangle$ to $\mathcal{C}$ via (8), Lemma 2 bounds the effect that the adversaries can have on $\tilde{M}_{\mathcal{C}}$. It is therefore the key technical tool powering our results, and is proved in Section A.4. Proposition 1 is proved from Lemmas 1 and 2 in Section A.5.

## 5 Recovering $T$ (Algorithm 2)

In this section we show that if $\tilde{M}$ satisfies the conclusion of Proposition 1, then Algorithm 2 recovers a set $T$ that approximates $T^*$ well. We represent the sets $T$ and $T^*$ as $\{0, 1\}$-vectors, and use the notation $\langle T, r \rangle$ to denote $\sum_{j \in [m]} T_j r_j$. Formally, we show the following:

**Proposition 2.** *Suppose Assumption 1 holds. For some $k_0 = \mathcal{O}\left( \frac{\log(2/\alpha \beta \epsilon \delta)}{\beta \epsilon^2} \right)$, with probability $1 - \delta$, Algorithm 2 outputs a set $T$ satisfying*

$$\langle T^* - T, r^* \rangle \leq \frac{2}{|\mathcal{C}|} \left( \sum_{i \in \mathcal{C}} \langle T^* - \tilde{M}_i, r^* \rangle \right) + \epsilon \beta m. \tag{9}$$

To establish Proposition 2, first note that with probability $1 - \frac{\delta}{2}$, at least one of the $\frac{2 \log(2/\delta)}{\alpha}$ randomly selected $i$ from Algorithm 2 will have cost $\langle T^* - \tilde{M}_i, r^* \rangle$ within twice the average cost across $i \in \mathcal{C}$. This is because with probability $\alpha$, a randomly selected $i$ will lie in $\mathcal{C}$, and with probability $\frac{1}{2}$, an $i \in \mathcal{C}$ will have cost at most twice the average cost (by Markov's inequality).

The remainder of the proof hinges on two results. First, we establish a concentration bound showing that $\sum_j \tilde{M}_{ij} \tilde{r}_j$ is close to $\frac{k_0}{m} \sum_j \tilde{M}_{ij} r_j^*$ for any fixed $i$, and hence (by a union bound) also for the $\frac{2 \log(2/\delta)}{\alpha}$ randomly selected $i$. This yields the following lemma, which is a straightforward application of Bernstein's inequality (see Section A.6 for a proof):

**Lemma 3.** *Let $i^*$ be the row selected in Algorithm 2. Suppose that $\tilde{r}$ satisfies Assumption 1. For some $k_0 = \mathcal{O}\left(\frac{\log(2/\alpha\delta)}{\beta\epsilon^2}\right)$, with probability $1 - \delta$, we have*

$$\langle T^* - \tilde{M}_{i^*}, r^* \rangle \leq \frac{2}{|\mathcal{C}|}\Big(\sum_{i \in \mathcal{C}} \langle T^* - \tilde{M}_i, r^* \rangle\Big) + \frac{\epsilon}{4}\beta m. \qquad (10)$$

Having recovered a good row $T_0 = \tilde{M}_{i^*}$, we need to turn $T_0$ into a binary vector so that Algorithm 2 can output a set; we do so via randomized rounding, obtaining a vector $T \in \{0,1\}^m$ such that $\mathbf{E}[T] = T_0$ (where the randomness is with respect to the choices made by the algorithm). Our rounding procedure is given in Algorithm 4; the following lemma, proved in A.7, asserts its correctness:

**Lemma 4.** *The output $T$ of Algorithm 4 satisfies $\mathbf{E}[T] = T_0$, $\|T\|_0 \leq \beta m$.*

---

**Algorithm 4** Randomized rounding algorithm.

1: **procedure** RANDOMIZEDROUND($T_0$)         ▷ $T_0 \in [0,1]^m$ satisfies $\|T_0\|_1 \leq \beta m$
2:      Let $s$ be the vector of partial sums of $T_0$        ▷ i.e., $s_j = (T_0)_1 + \cdots + (T_0)_j$
3:      Sample $u \sim \text{Uniform}([0,1])$.
4:      $T \leftarrow [0, \ldots, 0] \in \mathbf{R}^m$
5:      **for** $z = 0$ **to** $\beta m - 1$ **do**
6:          Find $j$ such that $u + z \in [s_{j-1}, s_j)$, and set $T_j = 1$.  ▷ if no such $j$ exists, skip this step
7:      **end for**
8:      **return** $T$
9: **end procedure**

---

The remainder of the proof involves lower-bounding the probability that $T$ is accepted in each stage of the while loop in Algorithm 2. We refer the reader to Section A.8 for details.

# 6  Open Directions and Related Work

**Future Directions.** On the theoretical side, perhaps the most immediate open question is whether it is possible to improve the dependence of $k$ (the amount of work required per worker) on the parameters $\alpha$, $\beta$, and $\epsilon$. It is tempting to hope that when $m = n$ a tight result would have $k = \tilde{\mathcal{O}}\left(\frac{1}{\alpha\beta\epsilon^2}\right)$, in loose analogy to recent results for the stochastic block model (Abbe and Sandon, 2015b;a; Banks and Moore, 2016). For stochastic block models, there is conjectured to be a gap between computational and information-theoretic thresholds, and it would be interesting to see if a similar phenomenon holds here (the scaling for $k$ given above is based on the conjectured computational threshold).

A second open question concerns the scaling in $n$: if $n \gg m$, can we get by with much less work per rater? Finally, it would be interesting to consider adaptivity: if the choice of queries is based on previous worker ratings, can we reduce the amount of work?

**Related work.** Our setting is closely related to the problem of *peer prediction* (Miller et al., 2005), in which we wish to obtain truthful information from a population of raters by exploiting inter-rater agreement. While several mechanisms have been proposed for these tasks, they typically assume that rater accuracy is observable online (Resnick and Sami, 2007), that the dishonest raters are rational agents maximizing a payoff function (Dasgupta and Ghosh, 2013; Kamble et al., 2015; Shnayder et al., 2016), that the raters follow a simple statistical model (Karger et al., 2014; Zhang et al., 2014; Zhou et al., 2015), or some combination of the above (Shah and Zhou, 2015; Shah et al., 2015). Ghosh et al. (2011) allow $o(n)$ adversaries to behave arbitrarily but require the rest to be stochastic.

The work closest to ours is Christiano (2014; 2016), which studies online collaborative prediction in the presence of adversaries; roughly, when raters interact with an item they predict its quality and afterwards observe the actual quality; the goal is to minimize the number of incorrect predictions among the honest raters. This differs from our setting in that (i) the raters are trying to learn the item qualities as part of the task, and (ii) there is no requirement to induce a final global estimate of the high-quality items, which is necessary for estimating quantiles. It seems possible however that there are theoretical ties between this setting and ours, which would be interesting to explore.

**Acknowledgments.** JS was supported by a Fannie & John Hertz Foundation Fellowship, an NSF Graduate Research Fellowship, and a Future of Life Institute grant. GV was supported by NSF CAREER award CCF-1351108, a Sloan Foundation Research Fellowship, and a research grant from the Okawa Foundation. MC was supported by NSF grants CCF-1565581, CCF-1617577, CCF-1302518 and a Simons Investigator Award.

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
