[Supplementary Material 1]

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

# A  Deferred Proofs

## A.1  Matrix Concentration Bound of Le et al. (2015)

For ease of reference, here we state the matrix concentration bound from Le et al. (2015), which we make use of in the proofs below.

**Theorem 3** (Theorem 2.1 in Le et al. (2015)). *Given an $s \times s$ matrix $P$ with entries $P_{i,j} \in [0,1]$, and a random matrix $A$ with the properties that 1) each entry of $A$ is chosen independently, 2) $\mathbb{E}[A_{i,j}] = P_{i,j}$, and 3) $A_{i,j} \in [0,1]$, then for any $r \geq 1$, the following holds with probability at least $1 - s^{-r}$: let $d = s \cdot \max_{i,j} P_{i,j}$, and modify any subset of at most $10s/d$ rows and/or columns of $A$ by arbitrarily decreasing the value of nonzero elements of those rows or columns to form the matrix $A'$ with entries in $[0,1]$, then*

$$||A' - P||_{op} \leq Cr^{3/2}\left(\sqrt{d} + \sqrt{d'}\right),$$

*where $d'$ is the maximum $\ell_2$ norm of any row or column of $A'$, and $C$ is an absolute constant.*

Note: The proof of this theorem in Le et al. (2015) shows that the statement continues to hold in the slightly more general setting where the entries of $A$ are chosen independently according to random variables with bounded variance and sub-Gaussian tails, rather than just random variables restricted to the interval $[0,1]$.

## A.2  Details of Lipschitz Bound (Equation 5)

The proof essentially consists of matching up each value $r_j^*$, for $j \in T^*$, with a set of values $r_{j'}^*$, $j' \geq j$, where the corresponding $\tilde{M}_{i,j'}$ sum to 1; we can then invoke the condition (2). Unfortunately, expressing this idea formally is a bit notationally cumbersome.

Before we start, we observe that the Lipschitz condition (2) implies that, if $r_j^* \geq r_{j'}^*$, then $r_j^* - r_{j'}^* \leq L \cdot \left(A_{i,j}^* - A_{i,j'}^*\right) + \epsilon_0$. It is this form of (2) that we will make use of below.

Now, let $I_j = \mathbf{I}[j \in T^*]$, and without loss of generality suppose that the indices $j$ are such that $r_1^* \geq r_2^* \geq \cdots \geq r_m^*$. For a vector $v \in [0,1]^m$, define

$$h(\tau, v) \stackrel{\text{def}}{=} \inf\{j \mid \sum_{j'=1}^{j} v_{j'} \geq \tau\}, \tag{11}$$

where $h(\tau, v) = \infty$ if no such $j$ exists. We observe that for any vector $v \in [0,1]^m$, we have

$$\sum_{j \in [m]} v_j r_j^* = \int_0^\infty r_{h(\tau;v)}^* d\tau, \tag{12}$$

where we define $r_\infty^* = 0$ (note that the integrand is therefore 0 for any $\tau \geq \|v\|_1$). Hence, we have

$$\sum_{j \in T^*} r_j^* - \sum_{j \in [m]} \tilde{M}_{i,j} r_j^* = \sum_{j \in [m]} I_j r_j^* - \sum_{j \in [m]} \tilde{M}_{i,j} r_j^* \tag{13}$$

$$= \int_0^{\beta m} r_{h(\tau,I)}^* - r_{h(\tau,\tilde{M}_i)}^* d\tau \tag{14}$$

$$\stackrel{(i)}{\leq} \int_0^{\beta m} \left[L \cdot \left(A_{h(\tau,I)}^* - A_{h(\tau,\tilde{M}_i)}^*\right) + \epsilon_0\right] d\tau \tag{15}$$

$$= L \cdot \left(\sum_{j \in [m]} I_j A_j^* - \sum_{j \in [m]} \tilde{M}_{i,j} A_j^*\right) + \beta m \epsilon_0 \tag{16}$$

$$= L \cdot \left(\sum_{j \in T^*} A_j^* - \sum_{j \in [m]} \tilde{M}_{i,j} A_j^*\right) + \beta m \epsilon_0, \tag{17}$$

which implies (5). The key step is (i), which uses the fact that $h(\tau, I) \leq h(\tau, \tilde{M}_i)$ (because $I$ is maximally concentrated on the left-most indices of $[m]$), and hence $r_{h(\tau,I)}^* \geq r_{h(\tau,\tilde{M}_i)}^*$.

## A.3 Stability Under Noise (Proof of Lemma 1)

By Hölder's inequality, we have that $|\langle \tilde{A} - B, M \rangle| \leq \|\tilde{A} - B\|_{\text{op}} \|M\|_*$. We now leverage Theorem 3 to bound $\|\tilde{A} - B\|_{\text{op}}$. To apply the theorem, first note that from the construction of $\tilde{A}$ given in Algorithm 3, $\tilde{A}$ can be constructed by first having the raters rate each item independently with probability $k/m$ to form matrix $\tilde{A}^o$ and then removing ratings from the "heavy" rows (i.e. rows with more than $2k$ ratings), and "heavy" columns (i.e. columns with more than $2k$ ratings). By standard Chernoff bounds, the probability that a given row or column will need to be "pruned" is at most $e^{-k/3} \leq 2/k$, and hence from the independence of the rows, the probability that more than $5n/k$ rows are "heavy" is at most $e^{-2n/3k}$. The probability that there are more than $5n/k$ heavy columns is identically bounded.

Note that the expectation of the portion of $\tilde{A}^o$ corresponding to the reliable raters is exactly the corresponding portion of matrix $B$, and with probability at least $1 - 2e^{-2n/3k}$, at most $10n/k$ rows and/or columns of $\tilde{A}^o$ are pruned to form $\tilde{A}$. Consider padding matrices $\tilde{A}$ and $B$ with zeros, to form the $n \times n$ matrices $\tilde{A}'$ and $B'$. With probability $1 - 2e^{-2n/3k}$ the conditions of Theorem 3 now apply to $\tilde{A}'$ and $B'$, with the parameters $d = \frac{nk}{m} \leq k$, and $d' = 2k$. Hence for any $r \geq 1$, with probability at least $1 - 2e^{-2n/3k} - n^{-r}$

$$\|\tilde{A} - B\|_{\text{op}} = \|\tilde{A}' - B'\|_{\text{op}} \leq Cr^{3/2}\sqrt{k},$$

for some absolute constant $C$.

By assumption, $\|\tilde{M}\|_* \leq \frac{2}{\alpha\epsilon}\sqrt{\alpha\beta nm}$ and $\|M^*\|_* \leq \frac{2}{\alpha\epsilon}\sqrt{\alpha\beta nm}$. Hence setting $r = \log(1/\delta)$, and $k \geq C' \log^3(\frac{1}{\delta}) \frac{m/n}{\epsilon^4 \alpha^3 \beta}$ for some absolute constant $C'$, we have that with probability at least $1 - \delta$, we have

$$|\langle \tilde{A} - B, \tilde{M} \rangle| \leq \frac{1}{2}\epsilon\alpha\beta kn,$$

and $|\langle \tilde{A} - B, M^* \rangle|$ is bounded identically.

To conclude, we have the following:

$$\langle B, \tilde{M} \rangle \geq \langle \tilde{A}, \tilde{M} \rangle - \frac{1}{2}\epsilon\alpha\beta kn \tag{18}$$

$$\geq \langle \tilde{A}, M^* \rangle - \frac{1}{2}\epsilon\alpha\beta kn \text{ (since } \tilde{M} \text{ is optimal for } \tilde{A}) \tag{19}$$

$$\geq \langle B, M^* \rangle - \epsilon\alpha\beta kn, \tag{20}$$

which completes the proof.

## A.4 Bounding the Effect of Adversaries (Proof of Lemma 2)

In this section we prove Lemma 2. Let $\mathcal{P}_0$ be the superset of $\mathcal{P}$ where we have removed the nuclear norm constraint. By Lagrangian duality we know that there is some $\mu$ such that maximizing $\langle B, M \rangle$ over $\mathcal{P} \cap \{M_{\mathcal{C}} = R_{\mathcal{C}}\}$ is equivalent to maximizing $f_\mu(M) \stackrel{\text{def}}{=} \langle B, M \rangle + \mu\left(\frac{2}{\epsilon\alpha}\sqrt{\alpha\beta nm} - \|M\|_*\right)$ over $\mathcal{P}_0 \cap \{M_{\mathcal{C}} = R_{\mathcal{C}}\}$.

We start by bounding $\mu$. We claim that $\mu \leq \epsilon k\sqrt{\alpha\beta n/m}$. To show this, we will first show that $\langle B, M \rangle$ cannot get too large. Let $\mathcal{E}$ be the set of $(i, j)$ for which ratings are observed, and define the matrix $B'$ as $(B')_{ij} = \frac{k}{m} + \mathbf{I}[(i, j) \in \mathcal{E}](B_{ij} - 1)$; note that $(B - B')_{ij} = \mathbf{I}[(i, j) \in \mathcal{E}] - \frac{k}{m}$. For any $M \in \mathcal{P}_0$, we have, with probability $1 - \delta$ (over the randomness in $\tilde{A}$)

$$\langle B, M \rangle \leq \langle B', M \rangle + \langle B - B', M \rangle \tag{21}$$

$$\leq \beta kn + \|B - B'\|_{\text{op}}\|M\|_* \tag{22}$$

$$\stackrel{(i)}{\leq} \beta kn + \log^{3/2}(1/\delta) \cdot 2\sqrt{2k}\|M\|_* \tag{23}$$

$$\stackrel{(ii)}{\leq} k\left(\beta n + \frac{\epsilon\sqrt{\alpha\beta n/m}}{2}\|M\|_*\right). \tag{24}$$

For (ii) to hold, we need to take $k$ sufficiently large, but there is some $k = \mathcal{O}\left(\frac{\log^3(2/\delta)}{\alpha\beta\epsilon^2}\frac{m}{n}\right)$ that suffices. In (i) we used the matrix concentration inequality of Theorem 3, in a similar manner as was used in our proof of Lemma 1. Specifically, we consider padding $B$ and $B'$ with zeros so as to make both into $n \times n$ matrices. Provided the total number of raters and items whose initial assignments are removed in the second and third steps of the rater/item assignment procedure (Algorithm 3) is bounded by $10n/k$, which occurs with probability at least $1 - \delta/2$ given our choice of $k$, then Theorem 3 applies with $r = \log(1/\delta)$, and $d$ and $d'$ bounded by $2k$, yielding an operator norm bound of $r^{3/2}(\sqrt{k} + \sqrt{2k}) \le \log^{3/2}(1/\delta) \cdot 2\sqrt{2k}$, that holds with probability $1 - n^{-r} > 1 - \delta/2$.

Now, suppose that we take $\mu_0 = \epsilon\sqrt{\alpha\beta n/mk}$ and optimize $\langle B, M \rangle - \mu_0 \|M\|_*$ over $\mathcal{P}_0 \cap \{M_{\mathcal{C}} = R_{\mathcal{C}}\}$. By the above inequalities, we have $\langle B, M \rangle - \mu_0\|M\|_* \le \beta kn - \frac{\epsilon\sqrt{\alpha\beta n/mk}}{2}\|M\|_*$, and so any $M$ with $\|M\|_* > \frac{2}{\epsilon\alpha}\sqrt{\alpha\beta nm}$ cannot possibly be optimal, since the solution $M = 0$ would be better. Hence, $\mu_0$ is a large enough Lagrange multiplier to ensure that $M \in \mathcal{P}$, and so $\mu \le \mu_0 = \epsilon k\sqrt{\alpha\beta n/m}$, as claimed.

We next characterize the subgradient of $f_\mu$ at $M = M^*$. Define the projection matrix $P$ as

$$P_{i,i'} = \left\{ \begin{array}{ll} \frac{1}{|\mathcal{C}|} & : \quad i, i' \in \mathcal{C} \\ \delta_{i,i'} & : \text{else} \end{array} \right. . \tag{25}$$

Thus $PM = M$ if and only if all rows in $\mathcal{C}$ are equal to each other. In particular, $PM = M$ whenever $M_{\mathcal{C}} = R_{\mathcal{C}}$. Now, since $M^*$ is the maximum of $f_\mu(M)$ over all $M \in \mathcal{P}_0 \cap \{M_{\mathcal{C}} = R_{\mathcal{C}}\}$, there must be some $G \in \partial f_\mu(M^*)$ such that $\langle G, M - M^* \rangle \le 0$ for all $M \in \mathcal{P}_0 \cap \{M_{\mathcal{C}} = R_{\mathcal{C}}\}$. The following lemma says that without loss of generality we can assume that $PG = G$:

**Lemma 5.** *Suppose that $G \in \partial f(M^*)$ satisfies $\langle G, M - M^* \rangle \le 0$ for all $M \in \mathcal{P}_0 \cap \{M_{\mathcal{C}} = R_{\mathcal{C}}\}$. Then, $PG$ satisfies the same property, and lies in $\partial f(M^*)$ as well.*

We can further note (by differentiating $f_\mu$) that $G = B - \mu Z_0$, where $\|Z_0\|_{\text{op}} \le 1$[1]. Then $PG = PB - \mu PZ_0 = B - \mu PZ_0$. Let $r(M)$ denote the matrix where $M_{\mathcal{C}}$ is replaced with $R_{\mathcal{C}}$ (so $r(M) \in \mathcal{P}_0 \cap \{R_{\mathcal{C}} = M_{\mathcal{C}}\}$ whenever $M \in \mathcal{P}_0$). The rest of the proof is basically algebra; for any $M \in \mathcal{P}$, we have

$$\langle B, M - M^* \rangle \overset{(i)}{\le} f_\mu(M) - f_\mu(M^*) \tag{26}$$

$$\overset{(ii)}{\le} \langle B - \mu PZ_0, M - M^* \rangle \tag{27}$$

$$= \langle B - \mu PZ_0, M - r(M) \rangle + \langle B - \mu PZ_0, r(M) - M^* \rangle \tag{28}$$

$$\overset{(iii)}{\le} \langle B - \mu PZ_0, M - r(M) \rangle \tag{29}$$

$$\overset{(iv)}{=} \langle B_{\mathcal{C}} - \mu(PZ_0)_{\mathcal{C}}, M_{\mathcal{C}} - r(M)_{\mathcal{C}} \rangle \tag{30}$$

$$= \langle B_{\mathcal{C}} - \mu(PZ_0)_{\mathcal{C}}, M_{\mathcal{C}} - M_{\mathcal{C}}^* \rangle, \tag{31}$$

where (i) is by complementary slackness (either $\mu = 0$ or $\|M^*\|_* = \frac{2}{\alpha\epsilon}\sqrt{\alpha\beta nm}$); (ii) is concavity of $f_\mu$, and the fact that $B - \mu PZ_0$ is a subgradient; (iii) is the property from Lemma 5 ($\langle B - \mu PZ_0, r(M) - M^* \rangle \le 0$ since $r(M) \in \mathcal{P}_0$); and (iv) is because $M$ and $r(M)$ only differ on $\mathcal{C}$.

To finish, we will take $Z = \mu(PZ_0)_{\mathcal{C}}$. We note that $\|Z\|_{\text{op}} = \|\mu(PZ_0)_{\mathcal{C}}\|_{\text{op}} \le \mu\|PZ_0\|_{\text{op}} \le \mu\|Z_0\|_{\text{op}} \le \mu$. Moreover, $Z$ has rank 1 and so $\|Z\|_F = \|Z\|_{\text{op}} \le \mu \le \epsilon k\sqrt{\alpha\beta n/m}$, as was to be shown.

**Proof of Lemma 5.** First, since $PM = M$ for all $M \in \mathcal{P}_0 \cap \{M_{\mathcal{C}} = R_{\mathcal{C}}\}$, and $PM^* = M^*$, we have $\langle PG, M - M^* \rangle = \langle G, P(M - M^*) \rangle = \langle G, M - M^* \rangle \le 0$. We thus only need to show that

$PG$ is still a subgradient of $f_\mu$. Indeed, we have (for arbitrary $M$)

$$\langle PG, M - M^* \rangle = \langle G, PM - M^* \rangle \tag{32}$$

$$\overset{(i)}{\geq} f_\mu(PM) - f_\mu(M^*) \tag{33}$$

$$= \langle B, PM \rangle - \mu\|PM\|_* - f_\mu(M^*) \tag{34}$$

$$= \langle B, M \rangle - \mu\|PM\|_* - f_\mu(M^*) \tag{35}$$

$$\overset{(ii)}{\geq} \langle B, M \rangle - \mu\|M\|_* - f_\mu(M^*) \tag{36}$$

$$= f_\mu(M) - f_\mu(M^*), \tag{37}$$

where (i) is because $G \in \partial f_\mu(M^*)$, and (ii) is because projecting decreases the nuclear norm. Since the inequality $\langle PG, M - M^* \rangle \geq f_\mu(M) - f_\mu(M^*)$ is the defining property for $PG$ to lie in $\partial f_\mu(M^*)$, the proof is complete.

### A.5   Proof of Proposition 1

In this section, we will prove Proposition 1 from Lemmas 1 and 2. We start by plugging in $\tilde{M}$ for $M$ in Lemma 2. This yields $\langle B_{\mathcal{C}} - Z_{\mathcal{C}}, M_{\mathcal{C}}^* - \tilde{M}_{\mathcal{C}} \rangle \leq \langle B, M^* - \tilde{M} \rangle \leq \epsilon\alpha\beta kn$ by Lemma 1. On the other hand, we have

$$|\langle Z_{\mathcal{C}}, M_{\mathcal{C}}^* - \tilde{M}_{\mathcal{C}} \rangle| \leq \|Z_{\mathcal{C}}\|_F \|M_{\mathcal{C}}^* - \tilde{M}_{\mathcal{C}}\|_F \tag{38}$$

$$\leq \epsilon\sqrt{\alpha\beta nk/m}\sqrt{\|M_{\mathcal{C}}^* - \tilde{M}_{\mathcal{C}}\|_1 \|M_{\mathcal{C}}^* - \tilde{M}_{\mathcal{C}}\|_\infty} \tag{39}$$

$$\leq \epsilon\sqrt{\alpha\beta nk/m}\sqrt{2\alpha\beta mn} = \sqrt{2}\epsilon\alpha\beta kn. \tag{40}$$

Putting these together, we obtain $\langle B_{\mathcal{C}}, M_{\mathcal{C}}^* - \tilde{M}_{\mathcal{C}} \rangle \leq (1 + \sqrt{2})\epsilon\alpha\beta kn$. Expanding $\langle B_{\mathcal{C}}, M_{\mathcal{C}}^* - \tilde{M}_{\mathcal{C}} \rangle$ as $\frac{k}{m}\sum_{i\in\mathcal{C}}\left(\sum_{j\in[m]}(R_{ij} - \tilde{M}_{ij})A_{ij}^*\right)$, we obtain

$$\frac{1}{|\mathcal{C}|}\frac{1}{\beta m}\sum_{i\in\mathcal{C}}\sum_{j\in[m]}(T_j^* - \tilde{M}_{ij})A_{ij}^* \leq (1 + \sqrt{2})\epsilon. \tag{41}$$

Scaling $\epsilon$ by a factor of $1 + \sqrt{2}$ yields the desired result.

### A.6   Concentration Bounds for $\tilde{r}$ (Proof of Lemma 3)

First, let $I$ be the set of $\frac{2\log(2/\delta)}{\alpha}$ indices that are randomly selected in Algorithm 2. We claim that with probability $1 - \frac{\delta}{2}$ (over the choice of $I$), $\min_{i\in I}\langle T^* - \tilde{M}_i, r^* \rangle \leq \frac{2}{|\mathcal{C}|}\sum_{i\in\mathcal{C}}\langle T^* - \tilde{M}_i, r^* \rangle$. Indeed, the probability that this is true for a single element of $I$ is at least $\frac{\alpha}{2}$ by Markov's inequality (probability $\alpha$ that $i \in \mathcal{C}$, and probability at least $\frac{1}{2}$ that the inequality holds conditioned on $i \in \mathcal{C}$). Therefore, the probability that it is false for all $i \in I$ is at most $\left(1 - \frac{\alpha}{2}\right)^{|I|} \leq \exp\left(-\frac{\alpha}{2}|I|\right) \leq \frac{\delta}{2}$.

Now, fix $I$ and consider the randomness in $\tilde{r}$. For any $i \in I$, we would like to bound $\langle \tilde{M}_i, \tilde{r} - \frac{k_0}{m}r^* \rangle = \sum_{j\in[m]}\tilde{M}_{ij}\left(\tilde{r}_j - \frac{k_0}{m}r_j^*\right)$. The quantity $\tilde{M}_{ij}\left(\tilde{r}_j - \frac{k_0}{m}r_j^*\right)$ is a zero-mean random variable bounded in $[0, 1]$, and has variance at most $\frac{k_0}{m}\tilde{M}_{ij}^2$. Therefore, by Bernstein's inequality, and the fact that $\sum_j \tilde{M}_{ij}^2 \leq \beta m$, we have

$$\mathbf{P}\left[|\langle \tilde{M}_i, \tilde{r} - \frac{k_0}{m}r^* \rangle| \geq t\right] \leq 2\exp\left(-\frac{t^2}{2\beta k_0 + \frac{2}{3}t}\right). \tag{42}$$

Setting $t = 2\max\left(\sqrt{2\beta k_0 \log(2/\delta_0)}, \frac{2}{3}\log(2/\delta_0)\right)$, we see that a given $i$ satisfies

$$\mathbf{P}\left[|\langle \tilde{M}_i, \tilde{r} - \frac{k_0}{m}r^* \rangle|\right] \leq 2\max\left(\sqrt{2\beta k_0 \log(2/\delta_0)}, \frac{2}{3}\log(2/\delta_0)\right) \tag{43}$$

$$= \mathcal{O}\left(\max\left(\sqrt{\beta k_0 \log(2/\delta_0)}, \log(2/\delta_0)\right)\right) \tag{44}$$

with probability $1 - \delta_0$. Union bounding over all $i \in I$ and setting $\delta_0 = \frac{\alpha\delta}{4\log(1/\delta)}$, we have that $\left|\langle \tilde{M}_i, \tilde{r} - \frac{k_0}{m}r^*\rangle\right| \leq \mathcal{O}\left(\max\left(\sqrt{\beta k_0 \log(\frac{2}{\alpha\delta})}, \log(\frac{2}{\alpha\delta})\right)\right)$ with probability $1 - \frac{\delta}{2}$. Multiplying through by $\frac{m}{k_0}$, we get that $\left|\langle \tilde{M}_i, \frac{m}{k_0}\tilde{r} - r^*\rangle\right| \leq \beta m \cdot \mathcal{O}\left(\max\left(t, t^2\right)\right)$ with probability $1 - \delta$, where $t = \sqrt{\frac{\log(\frac{2}{\alpha\delta})}{\beta k_0}}$. For some $k_0 = \mathcal{O}\left(\frac{\log(\frac{2}{\alpha\delta})}{\beta\epsilon^2}\right)$, we have that $\left|\langle \tilde{M}_i, \frac{m}{k_0}\tilde{r} - r^*\rangle\right| \leq \frac{1}{8}\epsilon\beta m$ for all $i \in I$. In particular, if $i_0$ is the element of $I$ that minimizes $\langle T^* - \tilde{M}_i, r^*\rangle$, then we have

$$\langle T^* - \tilde{M}_{i^*}, r^*\rangle \leq \langle T^* - \tilde{M}_{i^*}, \frac{m}{k_0}\tilde{r}\rangle + \frac{1}{8}\epsilon\beta m \tag{45}$$

$$\leq \langle T^* - \tilde{M}_{i_0}, \frac{m}{k_0}\tilde{r}\rangle + \frac{1}{8}\epsilon\beta m \tag{46}$$

$$\leq \langle T^* - \tilde{M}_{i_0}, r^*\rangle + \frac{1}{4}\epsilon\beta m \tag{47}$$

$$\leq \frac{2}{|\mathcal{C}|}\left(\sum_{i \in \mathcal{C}}\langle T^* - \tilde{M}_i, r^*\rangle\right) + \frac{1}{4}\epsilon\beta m, \tag{48}$$

as was to be shown.

## A.7 Correctness of Randomized Rounding (Proof of Lemma 4)

Our goal is to show that the output of Algorithm 4 satisfies $\mathbf{E}[T] = T_0$. First, observe that since $(T_0)_j \leq 1$ for all $j$, each interval $[s_{j-1}, s_j)$ has length at most 1, and so the for loop over $z$ never picks the same index $j$ twice. Moreover, the probability that $j$ is included in $T_0$ is exactly $s_j - s_{j-1} = (T_0)_j$. The result follows by linearity of expectation.

## A.8 Correctness of Algorithm 2 (Proof of Proposition 2)

First, we claim that with probability $1 - \delta$ (over the random choice of $T$), we will invoke `RandomizedRound` at most $\frac{4\log(1/\delta)}{\epsilon\beta}$ times. To see this, note that $\mathbf{E}[\langle T, \tilde{r}\rangle] = \langle T_0, \tilde{r}\rangle$, and $\langle T, \tilde{r}\rangle \in [0, k_0]$ almost surely. By Markov's inequality, the probability that $\langle T, \tilde{r}\rangle < \langle T_0, \tilde{r}\rangle - \frac{\epsilon}{4}\beta k_0$ is at most $\frac{k_0 - \langle T_0, \tilde{r}\rangle}{k_0 - \langle T_0, \tilde{r}\rangle + (\epsilon/4)\beta k_0}$. We can assume that $\langle T_0, \tilde{r}\rangle \geq (\epsilon/4)\beta k_0$ (since otherwise we accept $T$ with probability 1), in which case the preceding expression is bounded by $\frac{k_0 - (\epsilon/4)\beta k_0}{k_0} = 1 - \frac{\epsilon}{4}\beta$. Therefore, the probability of accepting $T$ in any given iteration of the while loop is at least $\frac{\epsilon}{4}\beta$, and so the probability of accepting at least once in $\frac{4\log(1/\delta)}{\epsilon\beta}$ iterations is indeed at least $1 - \delta$.

Next, for some $k_0 = \mathcal{O}\left(\frac{\log(2/\alpha\beta\epsilon\delta)}{\beta\epsilon^2}\right)$, we can make the probability that $|\langle T, \tilde{r} - \frac{k_0}{m}r^*\rangle| \geq \frac{\epsilon}{4}\beta k_0$ be at most $\frac{\alpha\beta\epsilon\delta}{16\log^2(2/\delta)}$, with respect to the randomness in $\tilde{r}$ (this follows from a standard Bernstein argument which we omit as it is essentially the same as the one in Lemma 3). Therefore, by union bounding over the $\frac{4\log(1/\delta)}{\epsilon\beta}$ possible $T$ and the $\frac{2\log(2/\delta)}{\alpha}$ possible values of $T_0$ (one for each possible $i^*$), with probability $1 - 2\delta$ we have $|\langle T, \tilde{r} - \frac{k_0}{m}r^*\rangle| \leq \frac{\epsilon}{4}\beta k_0$ for whichever $T$ we end up accepting, as well as for $T = T_0$.

Consequently, we have

$$\langle T, r^*\rangle \geq \frac{m}{k_0}\langle T, \tilde{r}\rangle - \frac{\epsilon}{4}\beta m \tag{49}$$

$$\geq \frac{m}{k_0}\langle T_0, \tilde{r}\rangle - \frac{2\epsilon}{4}\beta m \tag{50}$$

$$\geq \langle T_0, r^*\rangle - \frac{3\epsilon}{4}\beta m \tag{51}$$

$$\geq \langle \frac{1}{|\mathcal{C}|}\sum_{i \in \mathcal{C}}\tilde{M}_i, r^*\rangle - \epsilon\beta m, \tag{52}$$

where the final step is Lemma 3. By scaling down the failure probability $\delta$ by a constant (to account for the probability of failure at each step of the above argument), Proposition 2 follows.

### A.9 Proof of Theorem 2

By Proposition 1, for some $k = \mathcal{O}\left(\frac{\log^3(2/\delta)}{\beta\alpha^3\epsilon^4}\frac{m}{n}\right)$, we can with probability $1 - \delta$ recover a matrix $\tilde{M}$ such that $\frac{1}{|\mathcal{C}|}\frac{1}{\beta m}\sum_{i\in\mathcal{C}}\sum_{j\in[m]}(T_j^* - \tilde{M}_{ij})A_{ij}^* \leq \epsilon$, and hence by (5) $\tilde{M}$ also satisfies $\frac{1}{|\mathcal{C}|}\frac{1}{\beta m}\sum_{i\in\mathcal{C}}\sum_{j\in[m]}(T_j^* - \tilde{M}_{ij})r_j^* \leq L \cdot \epsilon + \epsilon_0$.

By Proposition 2, we then (with overall probability $1 - 2\delta$) recover a set $T$ satisfying

$$\frac{1}{\beta m}\langle T^* - T, r^*\rangle \leq \left(\frac{1}{\beta m}\frac{2}{|\mathcal{C}|}\sum_{i\in\mathcal{C}}\langle T^* - \tilde{M}_{ij}, r_j^*\rangle\right) + \epsilon \tag{53}$$

$$\leq 2 \cdot (L \cdot \epsilon + \epsilon_0) + \epsilon \tag{54}$$

$$= (2L + 1) \cdot \epsilon + 2\epsilon_0. \tag{55}$$

Theorem 2 follows by scaling down $\delta$ by a factor of 2.

## B   Examples of Adversarial Behavior

In this section, in order to provide some intuition we show two possible attacks that adversaries could employ to make it hard for us to recover the good items. The first attack creates a symmetric situation, whereby there are $\frac{1}{\alpha}$ indistinguishable sets of potentially good items, and we are therefore forced to consider each set before we can find out which one is actually good. The second attack demonstrates the necessity of constraining each row to have a fixed sum, by showing that adversaries that are allowed to create very dense rows can have disproportionate influence on nuclear norm-based recovery algorithms

### B.1   Necessity of Nuclear Norm Scaling

Suppose for simplicity that $\alpha = \beta$ and $n = m$. Let $J$ be the $\alpha n \times \alpha n$ all-ones matrix, and suppose that the full rating matrix $A$ has a block structure:

$$A^* = \begin{bmatrix} J & (1-\epsilon)J & \cdots & (1-\epsilon)J \\ (1-\epsilon)J & J & \cdots & (1-\epsilon)J \\ \vdots & \vdots & \ddots & \vdots \\ (1-\epsilon)J & (1-\epsilon)J & \cdots & J \end{bmatrix} \tag{56}$$

In other words, both the items and raters are partitioned into $\frac{1}{\alpha}$ blocks, each of size $\alpha n$. A rater assigns a rating of $1$ to everything in their corresponding block, and a rating of $1 - \epsilon$ to everything outside of their block. Thus, there are $\frac{1}{\alpha}$ completely symmetric blocks, only one of which corresponds to the good raters. Since we do not know which of these blocks is actually good, we need to include them all in our solution $M^*$. Therefore, $M^*$ should be

$$M^* = \begin{bmatrix} J & 0 & \cdots & 0 \\ 0 & J & \cdots & 0 \\ \vdots & \vdots & \ddots & \vdots \\ 0 & 0 & \cdots & J \end{bmatrix} \tag{57}$$

Note however that in this case, $\|M^*\|_* = n$, while $\sqrt{\alpha\beta nm} = \sqrt{\alpha^2 n^2} = \alpha n$. We therefore need the nuclear norm constraint in (1) to be at least $\frac{1}{\alpha}$ times larger than $\sqrt{\alpha\beta nm}$ in order to capture the solution $M^*$ above.

It is not obvious to us that the additional $\frac{2}{\epsilon}$ factor appearing in (1) is actually necessary, but it was needed in our analysis in order to bound the impact of adversaries.

### B.2   Necessity of Row Normalization

Suppose that we did not include the row-normalization constraint $\sum_j \tilde{M}_{ij} \leq \beta m$ in (1). For instance, this might happen if, instead of seeking all items of quality above a given quantile, we sought all

items with quality above a given *threshold* (say, whose quality was great than $\frac{1}{2}$). In this case we might pose the optimization problem

$$\text{maximize } \langle \tilde{A} - \tfrac{1}{2} J_{n,m}, M \rangle, \tag{58}$$
$$\text{subject to } 0 \leq M_{ij} \leq 1 \;\; \forall i, j,$$
$$\|M\|_* \leq \frac{2}{\alpha \epsilon} \sqrt{\alpha \beta nm},$$

where $J_{n,m}$ is the $n \times m$ all-ones matrix. There are several reasons not to do this (for instance, focusing on quality thresholds rather than quantile thresholds loses the robustness to monotonic transformations that our method enjoys). In this section, we will focus on the particular issue that (58) is *less robust to adversaries* than (1).

Concretely, we will suppose that the adversaries are split into $\frac{1}{3\beta}\left(\frac{1}{\alpha} - 1\right)$ blocks of size $3\alpha\beta n$, each of which rates a random subset of $\frac{m}{2}$ items positively and the rest negatively. So for instance the matrix $A^*$ might look like (with $\alpha = \frac{2}{5}, \beta = \frac{1}{6}, n = 10, m = 12$):

$$A^* = \begin{array}{c} \text{good} \\ \text{bad 1} \\ \text{bad 2} \\ \text{bad 3} \end{array} \begin{bmatrix} 1 & 1 & 0 & 0 & 0 & 0 & 0 & 0 & 0 & 0 & 0 & 0 \\ 1 & 1 & 0 & 0 & 0 & 0 & 0 & 0 & 0 & 0 & 0 & 0 \\ 1 & 1 & 0 & 0 & 0 & 0 & 0 & 0 & 0 & 0 & 0 & 0 \\ 1 & 1 & 0 & 0 & 0 & 0 & 0 & 0 & 0 & 0 & 0 & 0 \\ 0 & 1 & 0 & 0 & 1 & 0 & 0 & 0 & 1 & 1 & 1 & 1 \\ 0 & 1 & 0 & 0 & 1 & 0 & 0 & 0 & 1 & 1 & 1 & 1 \\ 0 & 1 & 1 & 0 & 1 & 1 & 0 & 1 & 0 & 0 & 1 & 0 \\ 0 & 1 & 1 & 0 & 1 & 1 & 0 & 1 & 0 & 0 & 1 & 0 \\ 1 & 0 & 1 & 1 & 0 & 0 & 1 & 0 & 0 & 1 & 0 & 1 \\ 1 & 0 & 1 & 1 & 0 & 0 & 1 & 0 & 0 & 1 & 0 & 1 \end{bmatrix} \tag{59}$$

The nuclear norm of each individual bad block is $\sqrt{\frac{3}{2}\alpha\beta nm}$, and because the blocks are chosen independently of each other, the nuclear norm will be approximately additive across blocks. In addition, including a given bad block increases $\langle \tilde{A} - \frac{1}{2}J, M \rangle$ by $\frac{3}{4}\alpha\beta nm$. In contrast, including the good block increases the nuclear norm by $\sqrt{\alpha\beta nm}$ and only increases the objective by $\frac{1}{2}\alpha\beta nm$. The bad blocks therefore all give more "bang for the buck" in terms of how much they increase the objective vs. how much much they increase the nuclear norm, so we will add them before the good block.

To accomodate all these bad blocks, we need to allow $\|M\|_*$ to be at least roughly $\frac{1}{3\beta}\left(\frac{1}{\alpha} - 1\right) \times \sqrt{\frac{3}{2}\alpha\beta nm} = \Omega\left(\frac{1}{\alpha\beta}\sqrt{\alpha\beta nm}\right)$, which is adds an extra factor of $\frac{1}{\beta}$ relative to when we constrain the column sum. The issue can be seen in the above construction in (59): if we do not normalize the rows, then the rows controlled by adversaries can exert disproportionate influence (up to a factor of $\frac{1}{\beta}$) by creating columns that are much denser than those of the reliable raters.

[Supplementary Material 2]

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

3: $\qquad$ Sample $u \sim \text{Uniform}([0,1])$.
4: $\qquad$ $T \leftarrow [0, \ldots, 0] \in \mathbf{R}^m$
5: $\qquad$ **for** $z = 0$ **to** $\beta m - 1$ **do**
6: $\qquad\qquad$ Find $j$ such that $u + z \in [s_{j-1}, s_j)$, and set $T_j = 1$.  ▷ if no such $j$ exists, skip this step
7: $\qquad$ **end for**
8: $\qquad$ **return** $T$
9: **end procedure**

---

The remainder of the proof involves lower-bounding the probability that $T$ is accepted in each stage of the while loop in Algorithm 2. We refer the reader to Section A.8 for details.

## 6 Open Directions and Related Work

**Future Directions.** On the theoretical side, perhaps the most immediate open question is whether it is possible to improve the dependence of $k$ (the amount of work required per worker) on the parameters $\alpha, \beta$, and $\epsilon$. It is tempting to hope that when $m = n$ a tight result would have $k = \tilde{\mathcal{O}}\left(\frac{1}{\alpha\beta\epsilon^2}\right)$, in loose analogy to recent results for the stochastic block model (Abbe and Sandon, 2015b;a; Banks and Moore, 2016). For stochastic block models, there is conjectured to be a gap between computational and information-theoretic thresholds, and it would be interesting to see if a similar phenomenon holds here (the scaling for $k$ given above is based on the conjectured computational threshold).

A second open question concerns the scaling in $n$: if $n \gg m$, can we get by with much less work per rater? Finally, it would be interesting to consider adaptivity: if the choice of queries is based on previous worker ratings, can we reduce the amount of work?

**Related work.** Our setting is closely related to the problem of *peer prediction* (Miller et al., 2005), in which we wish to obtain truthful information from a population of raters by exploiting inter-rater agreement. While several mechanisms have been proposed for these tasks, they typically assume that rater accuracy is observable online (Resnick and Sami, 2007), that the dishonest raters are rational agents maximizing a payoff function (Dasgupta and Ghosh, 2013; Kamble et al., 2015; Shnayder et al., 2016), that the raters follow a simple statistical model (Karger et al., 2014; Zhang et al., 2014; Zhou et al., 2015), or some combination of the above (Shah and Zhou, 2015; Shah et al., 2015). Ghosh et al. (2011) allow $o(n)$ adversaries to behave arbitrarily but require the rest to be stochastic.

The work closest to ours is Christiano (2014; 2016), which studies online collaborative prediction in the presence of adversaries; roughly, when raters interact with an item they predict its quality and afterwards observe the actual quality; the goal is to minimize the number of incorrect predictions among the honest raters. This differs from our setting in that (i) the raters are trying to learn the item qualities as part of the task, and (ii) there is no requirement to induce a final global estimate of the high-quality items, which is necessary for estimating quantiles. It seems possible however that there are theoretical ties between this setting and ours, which would be interesting to explore.

**Acknowledgments.** JS was supported by a Fannie & John Hertz Foundation Fellowship, an NSF Graduate Research Fellowship, and a Future of Life Institute grant. GV was supported by NSF CAREER award CCF-1351108, a Sloan Foundation Research Fellowship, and a research grant from the Okawa Foundation. MC was supported by NSF grants CCF-1565581, CCF-1617577, CCF-1302518 and a Simons Investigator Award.