[Reviews · NeurIPS 2016]

Reviewer 1

Summary

This paper considers the problem of aggregating the work of many crowdsourcing workers, each of whom is capable of providing ratings on a set of m available items. The goal is to have the workers provide ratings for only a small fraction of the items, and still recover the "top K" items (with limited error) despite the potential for adversarial workers who can provide false or damaging ratings to the system. The authors provide a set of algorithms in how to assign workers to tasks, and then from their responses how to estimate the highest-quality items. The main result of the paper is an associated bound that specifies how many ratings per worker are required to get a good estimate of the top group in terms of the error rate of the workers, the number of "top items" required, and the fraction of poor workers.

Qualitative Assessment

My overall evaluation of this work is that I find it quite interesting, but with some caveats and concerns. Here are the high-level points: 1. The goal of the paper, which is to design an algorithm to find highly-desired items in a large pool, seems very natural to me. I am also quite satisified with the worker model, which I think is reasonably realistic: some fraction of the workers provide accurate and high-quality ratings with few errors, whereas another fraction of the workers provide poor or potentially-harmful ratings, and the algorithm is able to obtain a small number of "gold standard ratings". Within this setting, the authors provide a very straightforward solution using a "matrix completion"-like algorithm, and they combine a number of interesting ideas (matrix concentration, nuclear-norm regularization, etc.) to obtain their result. The natural question to ask is "how many worker ratings and gold-standard ratings are needed?" and the bound proven by the authors provides a reasonably sufficient and clear answer. 2. One of the downsides of the paper is that it seems to be trying to do a lot in 8 pages. The paper proposes one main algorithm, but includes about 3-4 other subroutines, each of which involves some significant complexity and detail. There is an application of randomized rounding, the authors need to apply a Lagrangian duality argument, there is the use of a matrix optimization with a trace-norm constraint, etc. There is a lot of notation as well, and keeping it all straight while reading is quite a challenge. So one major objection is that the authors simply tried to do too much here, and maybe went a little overboard without describing in suitable detail all of the pieces. 3. A bigger concern is that the paper is sloppy in a number of places, and one has the impression that it was written in a hurry without a careful reading. Here are just a few (big) examples: -- In the definition of the setting, the authors refer to "when i \in [m] and j \in [m]" which suggests that the #workers and #items are the same (which is suggested in the intro as well). But later it's clear that the authors probably meant "i \in [n]" -- The description in Figure 1 says "indicated by \tilde{r}" but I don't see that character anywhere in the figure (and I wanted to understand what it means) -- The authors refer to "rating it ourself" -- who is "ourself"??! This was a very confusing way of describing what I can only assume to be a "gold standard oracle". -- I tried to look at references in the main body of the document to see what papers were being cited and... I found nothing. Where is Moitra et al? Agarwal et al? Mannor/Tsitsiklis? These missing references were very suspicious to me, and it makes it very hard to evaluate the novelty. -- This seems to be a highly-relevant paper which doesn't appear to have been cited: *Who Moderates the Moderators? Crowdsourcing Abuse Detection in User-Generated Content* A Ghosh, S Kale, P McAfee I would lean towards acceptance for this paper, as many of the issues above can be fixed with a good revision. But indeed some doubts still linger.

Confidence in this Review

2-Confident (read it all; understood it all reasonably well)


Reviewer 2

Summary

The paper considers a two-stage crowdsourcing setup, where one set of workers produce some output, and another (disjoint) set are tasked with assessing the quality of those outputs. The goal of the manager is to curate the best-quality outputs, as measured by ensuring the average quality is only epsilon worse than the top beta-fraction, while minimizing the number of outputs the manager must personally assess. An algorithm is given which achieves this with effort by each worker and the manager which does not scale with the number of outputs to be rated.

Qualitative Assessment

This strikes me as a hard problem to solve, and techniques employed seem to confirm this. I am not sold on the model, set-up, and formal curation goal, however; all seem somewhat ad-hoc. Moreover, the bounds, while not growing with n, still have somewhat formiddable dependencies on the relevant quantities -- in particular, each worker must rate Omega(1/beta alpha^3 epsilon^4), which seems impractical even if all constants are 1/2. It was not clear to me how this work could apply to peer prediction, where the manager cannot personally examine any of the tasks. Please describe explicitly and prominently, or remove this claim (and from the title). 53: who is "we" here? 265: a more accurate characterization of this body of work would be that workers are assumed to either be honest or rational (i.e. if workers deviate, they do so in a way to maximize their expected payment-minus-effort)

Confidence in this Review

1-Less confident (might not have understood significant parts)


Reviewer 3

Summary

The authors propose an algorithm that detects your favourite items among many by detecting among a crowd of raters the ones that are the closest to your own taste. The algorithm uses sparse matrix completion to identify the raters that have common interest with yours. The rest of the raters can actually behave adversarialy. The analysis provides a sufficient condition in the number of reliable raters in the pool of raters and the amount of rating required so that you can find your favourite items.

Qualitative Assessment

- what is a node? line 74. - what does the following mean : 'in the sense of having lower expected accuracy on every item'? line 84 To me the explanation of the challenges of the problem at hand with respect to previous matrix completion/ clustering algorithm is unclear. (end of page 2 /beginning of page 3) - you ask for the rating of ko items. Isn't this close to using gold sets? - In theorem 1: the formula is not very clear on what is on the numerator and denominator. I am guessing that beta alpha and epsilon for instance are on the denominator. Am I right? - Why in this theorem is the result given with 99 percent probability. Why not giving it with 1-delta probability and make k and ko depend on delta? - Do you have lower bound that would give an idea of the gap and the potential improvement left. - line 61: you refer to Mannor and Tsitsiklis for an evaluation of epsilon accurate solution. Is there a result that refer to quantile in this reference. I could not find it. Can you please give a more precise reference. - how can you differentiate between a missing information (0) and a rating that is 0? - is it really surprising that k does not depend on n as the number of reliable raters already depends on n. minor comments: -------------- - 'the the' line 58 - ',via' line 61 : lack of space - in the caption of figure 1 : do you mean \tilde r or r^* ?

Confidence in this Review

1-Less confident (might not have understood significant parts)


Reviewer 4

Summary

The paper addresses a problem where crowdsourced data is obtained regarding evaluations of various items, and where the workers have a large proportion of adversarially behaved workers among them. The goal is to estimate the top certain fraction of items from this noisy data. The paper presents an algorithm and associated theoretical guarantees for this problem.

Qualitative Assessment

Overall I really like this paper. The problem is relevant, interesting and challenging. On the theoretical front, the paper is a solid piece of work. On the practical front though, the algorithm requires certain improvements to make practical impact. Constructive criticisms, comments and questions (I intend to read the answers in the rebuttal carefully): - The proposed algorithm (Algo. 1) requires that the values of alpha (fraction of adversarial workers) and epsilon (noise of the post hoc judgements) to be known, which limits the usability of the algorithm. - Ignoring the above point, was trying to come up with a concrete application for the setting/algorithm for the paper. The paper assumes that the items can be rated by themselves and correlation with their own answer is considered as a metric for "goodness" of a worker. This assumption rules out yelp, amazon etc. since the ratings there are highly subjective. The paper also addresses settings where a large number of workers are strategically adversarial. This rules out crowd labeling settings where one mostly finds spammers and little adversarial behavior. I would also rule out peer grading since (as we often observe in our classes) a strict negative penalty for "cheating" (which can be caught with the instructor grading some papers) may very well deter most adversarial behavior. - It is said that the result of allowing for a constant number of evaluations is 'surprising'. However, given that the problem formulation allows tolerance of a constant fraction of error, and that it assumes a constant fraction of adversarial workers, it is not very surprising that under this formulation that only a constant number of samples per worker are sufficient. As soon as the error tolerance \delta is allowed to scale with n (i.e., when one wants the probability of error to go to zero as n goes to infinity) then the constant-sample guarantee no longer holds. - The paper first suggests that "gold sets" may not be useful (lines 23-27), but then subsequently assume a setting where they have access to (almost) gold set data. Aren't the "post hoc" judgements identical to gold sets but with some controlled noise? On a related note, the introduction uses Yelp and Amazon ratings as motivating applications. However since the items on these platforms are all subjective, what does the "expert" post hoc judgement mean in this context? - Algo. 2: How is the "un assignment" done? Is it arbitrary? - Does the claim of Proposition 1 hold with probability 1 (as seems to be suggested by the statement of the proposition)? - At several points in the proofs, the probability space of operation is not clear. For instance, in line 442, both T and tilde{r'} are stochastic. The underlying probability space in the context of this line, is however, is not the joint (T, tilde{r'}) space but rather where it is conditioned on tilde{r'}; however, this is never mentioned in the proof. Such an issue occurs in several proofs, where the reader is required to read the proof ahead and reverse engineer the underlying probability space. - Should there be \tilde{M} in lines 458-459?

Confidence in this Review

3-Expert (read the paper in detail, know the area, quite certain of my opinion)


Reviewer 5

Summary

This paper provides a new algorithm in the context of crowdsourced ratings. In particular the method allows to recover a set of items that have the highest quality, s.t. the average quality is within an eps bound of the true best items with a probability of 99%. The algorithm makes minimal assumptions about the raters. In particular the set of 'bad raters' is unknown a priory and 'bad raters' are allowed to strategically collude in order to change results. Also, the algorithm can be applied in situations where a majority of raters is adversarial. Interestingly the algorithm requires 'high quality' ratings for a number of items that is independent of the total number of items considered. The method relies on two steps: First a matrix M_i,j is recovered from the sparse and noisy ratings matrix, which indicates whether the item i,j is in the top quantile of the ratings for this rater. Next the raters with the greatest amount of overlap with a few hand-rated items are selected in order to identify the best items.

Qualitative Assessment

The paper presents new results in the relevant area of crowdsourced quality prediction. In particular it provides a method that allows recovering the top beta elements in a set of n elements using a number of high quality ratings that is independent of the number of items being evaluated. However, the number of crowd reviews needed scales highly unfavourable with respect to the closeness of the recovered items and a ground truth, as O( 1 / eps^4 ). The main novelty lies in the generality of the problem setting: First of all the paper allows for any possible behaviour amongst the 'bad' reviewers (adversarial, random etc). Secondly the paper does not assume any 'golden set' of true labels. On the technical side a novel contribution is the proof of the bounds on adversarial impact due to the nuclear norm using Lagrangian multipliers. The proofs in the paper are clearly presented, but it would also be interesting to see a real world experimental application of the method as additional validation.

Confidence in this Review

2-Confident (read it all; understood it all reasonably well)


Reviewer 6

Summary

The papers proposes a new algorithm for a reliable evaluation of crowdsourced ratings. The algorithm is designed to make little assumptions about the raters, e.g. it does not assume that the majority of raters is reliable or that there is a 'gold set' data.

Qualitative Assessment

The paper is interesting, and its approach to relaxing assumptions on the rater's behavior could have a lasting impact on understanding quality of crowdsourced predictions. However, a real world application is necessary in order to demonstrate algorithms performance under natural conditions, and to confirm that the assumptions present in the paper still hold. One of the main assumptions is that you are not using a 'gold set' for the analysis. However, you are asking to rate a small number of items. How do these two statements relate to each other? page 3 - a lot of references are missing

Confidence in this Review

2-Confident (read it all; understood it all reasonably well)